# Neuroprotective Effects of an Edible Pigment Brilliant Blue FCF against Behavioral Abnormity in MCAO Rats

**DOI:** 10.3390/ph15081018

**Published:** 2022-08-18

**Authors:** Jingyang Le, Xiao Xiao, Difan Zhang, Yi Feng, Zhuoying Wu, Yuechun Mao, Chenye Mou, Yanfei Xie, Xiaowei Chen, Hao Liu, Wei Cui

**Affiliations:** Translational Medicine Center of Pain, Emotion and Cognition, Ningbo Key Laboratory of Behavioral Neuroscience, Zhejiang Provincial Key Laboratory of Pathophysiology, School of Medicine, Ningbo University, Ningbo 315211, China

**Keywords:** ischemic stroke, brilliant blue FCF, ERK, GSK3β, MCAO

## Abstract

Ischemic stroke leads to hypoxia-induced neuronal death and behavioral abnormity, and is a major cause of death in the modern society. However, the treatments of this disease are limited. Brilliant Blue FCF (BBF) is an edible pigment used in the food industry that with multiple aromatic rings and sulfonic acid groups in its structure. BBF and its derivatives were proved to cross the blood-brain barrier and have advantages on the therapy of neuropsychiatric diseases. In this study, BBF, but not its derivatives, significantly ameliorated chemical hypoxia-induced cell death in HT22 hippocampal neuronal cell line. Moreover, protective effects of BBF were attributed to the inhibition of the extracellular regulated protein kinase (ERK) and glycogen synthase kinase-3β (GSK3β) pathways as evidenced by Western blotting analysis and specific inhibitors. Furthermore, BBF significantly reduced neurological and behavioral abnormity, and decreased brain infarct volume and cerebral edema induced by middle cerebral artery occlusion/reperfusion (MCAO) in rats. MCAO-induced increase of p-ERK in ischemic penumbra was reduced by BBF in rats. These results suggested that BBF prevented chemical hypoxia-induced otoxicity and MCAO-induced behavioral abnormity via the inhibition of the ERK and GSK3β pathways, indicating the potential use of BBF for treating ischemic stroke

## 1. Introduction

Stroke is the leading cause of death in the modern society, while ischemic stroke accounts for the most cases of stroke worldwide [1]. During ischemic stroke, the blood flow is reduced after cerebral thrombosis, which further causes cerebral ischemia and hypoxia, leading to the irreversible loss of functional neurons and motor and neurological impairments [2]. Currently, only recombinant tissue plasminogen activator (rtPA) was approved by US Food and Drug Administration to treat ischemic stroke in the acute stage. However, the therapeutic window of rtPA is quite narrow, and rtPA might damage blood–brain barrier, and produce side effects, such as the increase of risk of cerebral edema and cerebral hemorrhage, leading to poor prognostic outcomes for patients [3,4]. Therefore, it is urgent to discover novel treatments for ischemic stroke.

Middle cerebral artery is the most common site of cerebral hemorrhage in humans. The transient occlusion of middle cerebral artery leads to ischemic injure in rodents, and the surgery of middle cerebral artery occlusion/reperfusion (MCAO) is a golden standard operation to establish an animal model of ischemic stroke [5]. Iodoacetic acid (IAA) is a 3-phosphate glyceraldehyde dehydrogenase inhibitor, and leads to the inhibition of energy metabolism, the depletion of intracellular ATP depletion, and the production of reactive oxygen species (ROS) in cells [6]. IAA-induced chemical hypoxia in HT22 hippocampal cell line could mimic neuronal injury during ischemic stroke in animals and was applied as an in vitro screening model for anti-stroke neuroprotective drugs [7].

The abnormal activation of mitogen-activated protein kinases (MAPKs) plays an important role in hypoxia neuronal death within ischemic penumbra during stroke. The extracellular regulated protein kinase (ERK) and glycogen synthase kinase-3β (GSK3β) are the main MAPKs, and were reported to be regulated during ischemic injury [8]. ERK is normally localized in the cytoplasm and transfers to the nucleus to enhance transcription factor activity. Many stimulations from growth factors snd cytokines, as well as viruses, increase the activation of ERK, and modulate the functions of neurons [9]. Pathological activation of ERK was found in the brain of patients with neurodegenerative disorders and traumatic brain injury [10]. Moreover, ERK was reported to be activated and phosphorylated during ischemic process, leading to the overexpression of pro-inflammatory cytokines and neuronal injury. In addition, the inhibition of the ERK pathway prevented ischemic injury in MCAO animals [11]. GSK3β is a multifunctional serine/threonine kinase acting on biological processes, such as neuronal apoptosis and neurotoxicity [12]. The blockade of GSK3β has been shown to produce neuroprotective effects against ischemic stroke [13]. Therefore, it is possible to inhibit ERK and GSK3β, concurrently, to treat ischemic stroke.

Many pigments produce neuroprotective effects and were used in the clinical trials to treat neurological disorders. For example, anthocyanins produce anti-oxidative stress effects, and has potential benefits for cognition, anxiety and depression [14]. Curcumin has been used in clinical trials in the treatment of neurogenerative disorders [15]. Brilliant Blue FCF (BBF) is an edible pigment used in the food industry that with multiple aromatic rings and sulfonic acid groups in its structure. BBF and its derivatives, including Brilliant Blue R (BBR), Brilliant Blue G (BBG), and Fast Green FCF (FGF), could readily cross the blood-brain barrier and have advantages to treat neuropsychiatric diseases [16]. Moreover, some of BBF derivatives decrease lipopolysaccharide- and β-amyloid-induced neurotoxicity in vitro, suggesting the neuroprotective potentials of BBF derivatives [16,17]. A previous study suggested that BBF and its derivatives potently inhibited the activity of MAPKs [18]. However, it is still unknown whether these compounds were able to produce anti-ischemia stroke neuroprotective effects.

In this study, we screened the most potent BBF derivatives from BBF, BBR, BBG, and FGF with the purpose of producing anti-chemical hypoxia protective effects in vitro. We further investigated the underlying neuroprotective mechanisms of the selected compound, with the emphasis on the regulation of ERK and GSK3β cascades in MCAO rats.

## 2. Results

### 2.1. BBF but Not BBF Derivatives Prevents Chemical Hypoxia-Induced Cell Death in HT22 Cells

We have previously established chemical hypoxia model induced by IAA in HT22 cells. In this study, the IC_50_ value of IAA to inhibit cell viability was about 10 μM (Figure 1A). Therefore, we chose 10 μM IAA for the following experiments. HT22 cells were pre-treated with various concentrations (2.5–20 μM) of BBF and its derivatives for 0.5 h, followed by 10 μM IAA for another 24 h. 3(4,5-dimethylthiazol-2-yl)-2.5-diphenyltetrazolium bromide (MTT) assay was used to measure cell viability. BBF (10–20 μM) significantly prevented IAA-induced reduction of cell viability (*p* < 0.05, one-way ANOVA and Tukey’s test, Figure 1B). However, BBR, BBG, and FGF did not exert protective effects in the same model.

The release of lactate dehydrogenase (LDH) was measured to evaluate the changes of cell membrane permeability. BBF (5–10 μM) significantly prevented the increase of LDH release induced by IAA (*p* < 0.05, one-way ANOVA and Tukey’s test, Figure 1C). Furthermore, BBF at 10 μM significantly prevented toxicity induced by IAA as illustrated by Fluorescein diacetate (FDA)/propidium iodide (PI) double staining (*p* < 0.01, one-way ANOVA and Tukey’s test, Figure 1D,E).

Furthermore, cells were examined by flow cytometry using Annexin V-FITC and PI staining. BBF largely prevented IAA-increased percentage of apoptotic cells and IAA-decreased percentage of live cells (Figure 1F). These results demonstrated that BBF produced anti-chemical hypoxia protective effects in HT22 hippocampal cell line.

### 2.2. BBF Produces Anti-Chemical Hypoxia Protective Effects via the Inhibition of ERK and GSK3β Concurrently in HT22 Cells

The ERK and GSK3β signaling pathways were involved in ischemic stroke. Western blotting analysis was used to explore whether these molecules were involved in the protective effects of BBF in vitro. As shown in Figure 2A,B, the expression of p-ERK in IAA group was significantly higher than that in the control group (*p* < 0.01, one-way ANOVA and Tukey’s test). Moreover, the expression of p-ERK was significantly decreased after the treatment with BBF (*p* < 0.01, one-way ANOVA and Tukey’s test, Figure 2B). Furthermore, the expression of pSer9-GSK3β in the IAA group was significantly lower than that in the control group (*p* < 0.01, one-way ANOVA and Tukey’s test, Figure 2C). Pre-treatment of BBF increased the expression of pSer9-GSK3β in IAA-treated HT22 cells (Figure 2C).

SB415286 is a specific inhibitor of GSK3β, and U0126 is a specific inhibitor of MEK. Both SB415286 and U0126 significantly attenuated the toxicity of IAA in HT22 cells, indicating that the activation of ERK and GSK3β might be involved in toxicity induced by chemical hypoxia in vitro (*p* < 0.05, one-way ANOVA and Tukey’s test, Figure 2D). LY294002 is a phosphatidylinositol 3-kinase (PI3-K) specific inhibitor and inhibits the activation of GSK3β. In our study, 10 μM LY294002 significantly reversed the protective effects of BBF, further demonstrating that BBF exerts protective activity by inhibiting GSK3β (*p* < 0.05, one-way ANOVA and Tukey’s test, Figure 2E).

### 2.3. BBF Prevents Neurological and Behavioral Abnormity, Brain Infarct and ERK Activation in MCAO Rats

MCAO induces behavioral abnormity and brain infarct in rats. Edaravone is an antioxidant clinically used for treating ischemic stroke and was used as a positive control in our study [19]. The neurological De Ryck’s behavioral and beam working scores were significantly different among various groups (for neurological scores, F(4, 25) = 11.82, *p* < 0.01; for De Ryck’s behavioral scores, F(4, 25) = 30.84, *p* < 0.01; for beam working behavioral scores, F(4, 25) = 11.07, *p* < 0.01, one-way ANOVA, Figure 3A–C). MCAO-treated rats exhibited certain degrees of neurological injury (*p* < 0.01, one-way ANOVA and Tukey’s test, Figure 3A) and behavioral ability (*p* < 0.01, one-way ANOVA and Tukey’s test, Figure 3B,C) when compared with rats in the sham group. The behavioral scores of BBF-injected rats were significantly increased when compared with MCAO rats (*p* < 0.05, one-way ANOVA and Tukey’s test, Figure 3B,C).

2,3,5-triphenyltetrazolium chloride (TTC) staining was used to analyze infarct volume. There are significant differences among various groups for brain infarct and total edema area (for brain infarct area, F(4, 25) = 10.81, *p* < 0.01; for total edema area, F(4, 25) = 14.68, *p* < 0.01, one-way ANOVA and Tukey’s test, Figure 3D–F). A significantly larger infarct volume was found in MCAO rats compared with sham-operated rats (*p* < 0.01 one-way ANOVA and Tukey’s test, Figure 3E). Moreover, compared with the surgery group, the infarct areas in the edaravone and BBF group was significantly decreased (*p* < 0.05, one-way ANOVA and Tukey’s test, Figure 3E). All these results suggested that BBF significantly prevented MCAO-induced brain infarct with similar efficacy as edaravone.

We explored whether ERK or GSK3β was involved in the neuroprotective effects of BBF in vivo. The expression of p-ERK in ischemic penumbra was significantly increased in the surgical group compared with the sham group, implying that MCAO might activate the ERK pathway in rats (*p* < 0.01, one-way ANOVA and Tukey’s test, Figure 3F,G). The expression of p-ERK was significantly decreased after BBF treatment (*p* < 0.05, one-way ANOVA and Tukey’s test, Figure 3H), indicating that BBF prevented the activation of ERK in MCAO rats. However, the expression of pSer9-GSK3β was not affected by BBF in MCAO rats (data not shown).

## 3. Discussion

In this study, BBF, but not its derivatives, concentration-dependently prevented chemical hypoxia-induced cell death in vitro, and inhibited ischemia stroke-induced behavioral abnormity in rats. The neuroprotective effects of BBF might be related to the inhibition of ERK and GSK3β (Figure 4).

A variety of food colorants have been reported to produce neuroprotective effects in neurological diseases. For example, erythrosine B inhibited the formation of amyloid fibrils, and reduce β-amyloid-induced toxicity [20]. Carotenoids and curcumin counteracted the neurological deficits via the inhibition of the excessive production of ROS [21]. BBF was reported to prevent the neurotoxicity induced by β-amyloid [22,23]. Moreover, BBF did not produce carcinogenic or toxic effects at very high concentrations [24,25]. Interestingly, BBF increased the longevity of drosophila, indicating the anti-aging effects of BBF in animals [26]. In this study, BBF was the only compound among its derivatives to prevent chemical hypoxia-induced cell death in HT22 cells, suggesting that this compound may be used in the treatment of hypoxia-related disorders.

The ERK and GSK pathways regulate hypoxia and ischemic stroke-induced neuronal injury both in vitro and in vivo. BBF significantly reduced the expression of p-ERK, and U0126 prevented IAA-induced cell death, demonstrating that BBF might produce protective effects at least partially via the inhibition of the ERK pathway. GSK-3β participates in a variety of neurophysiological activities, such as neuronal plasticity and neuronal apoptosis [27]. Many potential anti-stroke candidates, such as OCT4B-190 and Apelin 13, produced neuroprotective effects against ischemic stroke via inhibiting GSK-3β [28,29]. pSer9-GSK3β is an inactive form of GSK3β. In this study, BBF increased the expression of pSer9-GSK3β in IAA-treated HT22 cells. In addition, GSK3β inhibitor counteracted IAA-induced hypoxia injury in vitro. A PI3-K inhibitor abolished the protective activity of BBF in HT22 cells. All these results suggested that the protective effect of BBF might be related to the inhibition of GSK3β.

The neuroprotective effects of BBF against ischemia stroke were further investigated in vivo. Our findings that BBF prevented MCAO-induced motor dysfunction and infarct area, demonstrated that BBF produced neuroprotection against ischemia stroke. In this model, edaravone also prevented ischemia stroke-induced behavioral and neurological impairments, which is consistent to a previous study [30]. To further investigate the molecular mechanisms underlying the neuroprotective effects of BBF in vivo, the expression of p-ERK was evaluated in the ischemic penumbra. BBF decreased the expression of p-ERK in MCAO mice, suggesting that the ERK cascade was involved in the neuroprotective effects of BBF in vivo.

What is the upstream molecule of the ERK and the GSK3β cascades acted by BBF? P2X7 receptor, a subtype of P2X receptors, was reported to play an important role in ischemic stroke [31]. The over-expression of P2X7 receptor was discovered around the infarct core in the brain, activating the downstream ERK and GSK3β pathways, and leading to secondary neuronal injury. Moreover, P2X7 receptor was expressed in hippocampal neurons, and the activation of P2X7 receptor was reported to induce neuronal death in HT22 cells [32]. BBF derivatives were reported to directly inhibit the activation of P2X7 receptor [33]. Therefore, we speculated that BBF might prevent ischemic injury via the inhibition of P2X7 receptor (Figure 4). However, the confirmation of this speculation requires further experiments.

This study still has some limitations. The ERK-independent mechanisms were not explored with the regard of the neuroprotective effects of BBF. BBF was reported to inhibit the expression of Cx43 and reduce the ratio of Bax/Bcl-2 to produce anti-Parkinson’s neuroprotective effects [34]. Whether BBF prevented ischemic stroke-induced neuronal injury via acting on Cx43 is largely not known. In addition, BBF was proved to prevent MCAO-induced behavioral changes in rats, it is still not known whether post-ischemic stroke treatment of BBF on rats attenuated MCAO-induced neurological and behavioral impairments.

## 4. Materials and Methods

### 4.1. HT22 Cells Culture and Treatments

HT22 cells were obtained from Chinese Academy of Sciences (Shanghai, China), and cultured in high glucose modified Eagle’s medium (DMEM) containing 10% fetal bovine serum and penicillin (100 U/mL)/streptomycin (100 μg/mL). HT22 cells were cultured in an incubator at 37 °C and 5% CO_2_. The medium was replaced once every three days. All experiments were carried out 24 h after the seeding of cells. Before the experiments, the medium was changed to DMEM without fetal bovine serum or penicillin/streptomycin.

### 4.2. Measurement of Cell Viability

Cell viability was measured by the MTT assay. Briefly, after drug treatment for 24 h, 5 mg/mL MTT solution (10 μL) was added to each well. After incubating at 37 °C for 4 h, 100 μL solvating solution (0.01 N HCl, 10% SDS) was added, and incubated for 20 h. The absorbance of the samples was measured at a wavelength of 570 nm with 655 nm as a reference wavelength.

### 4.3. Measurement of LDH Release

The release of LDH was evaluated 24 h after drug treating. Cells were incubated with 2% (*v*/*v*) Triton X-100 in culture medium for 30 min to achieve maximum LDH release. Extracellular release of LDH was determined in the conditioned media collected from the culture dishes by using LDH assay kit (Roche Diagnostics, Indianapolis, IN, USA) according to manufacturer’s instructions. Briefly, 50 μL supernatants was added into 50 μL reaction buffer. After mixing at room temperature for 30 min, the release of LDH was measured at a wavelength of 490 nm with 655 nm as a reference wavelength.

### 4.4. FDA/PI Double Staining Assay

Briefly, after incubating with 10 μg/mL FDA and 5 μg/mL PI, cells were imaged by UV light microscopy and phase contrast microscopy. To quantitatively evaluate cell viability, the number of FDA-positive and PI-positive cells was counted, respectively, and the photos were taken from five random fields of each well. % of FDA-positive cells = [number of FDA-positive cells/(number of PI-positive cells + number of FDA-positive cells)] × 100%.

### 4.5. Western Blotting Analysis

Western blotting analysis was performed as previously described [35]. Protein lysates were separated on SDS-polyacrylamide gels and transferred onto polyvinyldifluoride membranes. Primary antibodies were used to detected proteins after membrane blocking (p-GSK3β, Cat No 5558; GSK3β, Cat No 12456; p-ERK, Cat No 4370 and ERK, Cat No 4695, Cell Signaling Technology, Danvers, MA, USA; β-actin, Cat No AF7018, Affinity Biosciences, Cincinnati, OH, USA). Signals were obtained after binding to HRP-conjugated secondary antibodies (Cat No sc-2357, Santa Cruz Biotechnology, Santa Cruz, CA, USA). Blots were developed using the enhanced chemiluminescence plus kit (Advansta, San Jose, CA, USA).

### 4.6. Animals and Surgery

Animal experiments were conducted following National Institutes of Health (NIH) Guide for Care and Use of Laboratory Animals (NIH Publications No. 80-23, revised 1996) and were approved by the Animal Ethics and Welfare Committee of Ningbo University (SYXK-2019-0005). Male Sprague Dawley (SD) rats weighing 250–300 g were purchased from Zhejiang academy of medical sciences (Hangzhou, Zhejiang, China). Rats were divided into five groups (six animals each group) randomly. Sham group: rats were subjected to sham surgery without MCAO and treated with vehicle. Surgical group: rats were subjected to MCAO and treated with vehicle. Surgery + 30 and 50 mg/kg BBF groups: rats were intraperitoneally injected with 30 and 50 mg/kg BBF, respectively, 30 min before, during and 30 min after MCAO surgery. Surgery + edaravone (positive control) group: rats were intraperitoneally injected with 6 mg/kg edaravone 30 min before, during and 30 min after MCAO surgery [36]. Intraperitoneally injection was used because BBF is poorly absorbed from the gastrointestinal tract and mainly excreted unchanged [37]. In addition, the timing for the treatment of ischemic stroke was limited, normally at the first several hours after the onset of the disease [38]. Therefore, rats were injected with drugs 30 min before, during and 30 min after MCAO surgery. The allowable daily intake dose of BBF was 10 mg/kg body weight/day. According to human effective dose (HED) formula, the dose of BBF at 61 mg/kg is equivalent to 10 mg/kg in human as a recommend ADI. Therefore, we selected the doses of 30 and 50 mg/kg, which were slightly lower than 61 mg/kg of BBF to treat rats. In the clinical study, edaravone was used at 1 mg/kg body weight/day with a good safety profile with no adverse effect. According to the HED formula, 6 mg/kg edaravone in rats is equivalent to 1 mg/kg edaravone in human. Therefore, edaravone at 6 mg/kg was selected to treat animals. The surgery of MCAO was performed as described previously [39]. For analgesic treatment, the rats were anesthetized with 2% pentobarbital (i.p. injection, 0.5 mL/250 g) before the surgery. If the rats showed signs of awakening, 2% pentobarbital (i.p. injection, 0.1 mL/250 g) was further added. And this analgesic procedure was normally used by many studies [40,41,42]. For body temperature control, the body temperature of animals was maintained at 37 ± 0.5 °C using a heating pad during MCAO surgery, which was commonly used by many studies [43,44,45]. Surgical incision of the rat neck was conducted to expose the neck vessels. The right common carotid artery and the right external carotid artery were ligated, and the right internal carotid artery was isolated. A 3-0 uncoated monofilament nylon suture with a rounded tip was inserted from the internal carotid artery to the right middle cerebral artery. The blood flow to the middle cerebral artery was blocked. After 1 h, reperfusion was performed by withdrawing the suture. Rats in Sham group were subjected to the same surgical procedures except that the monofilament nylon suture was not inserted into the internal carotid artery.

### 4.7. Neurological Tests

Longa’s neurological tests were used to measure the neurological deficits of rats 24 h after MCAO surgery. 0—no obvious neurological impairment; 1—could not entirely stretch contralateral forelimbs; 2—contralateral circling while walking; 3—contralateral fall over while walking; 4—cannot walk and reduce consciousness.

### 4.8. Behavioral Tests

De Ryck’s behavioral and beam walking tests were used to evaluate motor deficits in rats. De Ryck’s behavioral test is a sensitive test which has been widely applied in functional evaluation in stroke rat models. It is a 16-point scale with eight sub-tests, each of which had a score from 0 (maximum deficit) to 2 (no deficit). Among the eight tests, six tests were designed to examine the forelimb’s functions, while the other two were designed to evaluate the placing of the hind paw. Beam-walking test is a six-point scoring system: 0—unable to put the affected hind paw on the horizontal surface; 1—place the hind paw on the horizontal surface and maintain balance for at least 5 sec; 2—traverse the beam with dragging the affected hind paw; 3—traverse the beam and at least once put the affected hind paw on the horizontal surface; 4—cross the beam with over half steps slipped; 5—use the affected hind paw more than half its steps; 6—traverse the beam with below three foot slips.

### 4.9. TTC Staining

After behavioral tests, the animals were sacrificed. The whole brain was removed and brain slices were placed in 2% TTC solution and covered with tin foil in a 37 °C incubator for 30 min. The slices were photographed 24 h later.

### 4.10. Statistical Analysis

Results are expressed as mean ± SEM. Statistical analysis was performed using one-way ANOVA and post hoc Tukey’s test. *p* < 0.05 was considered as statistically significant.

## 5. Conclusions

In conclusion, BBF prevented IAA-induced toxicity in HT22 cells and MCAO-induced neurological and behavioral abnormity in rats, possibly via the inhibition of the ERK and GSK3β pathways, indicating the potential use of BBF for treating ischemic stroke.

## Figures and Tables

**Figure 1 pharmaceuticals-15-01018-f001:**
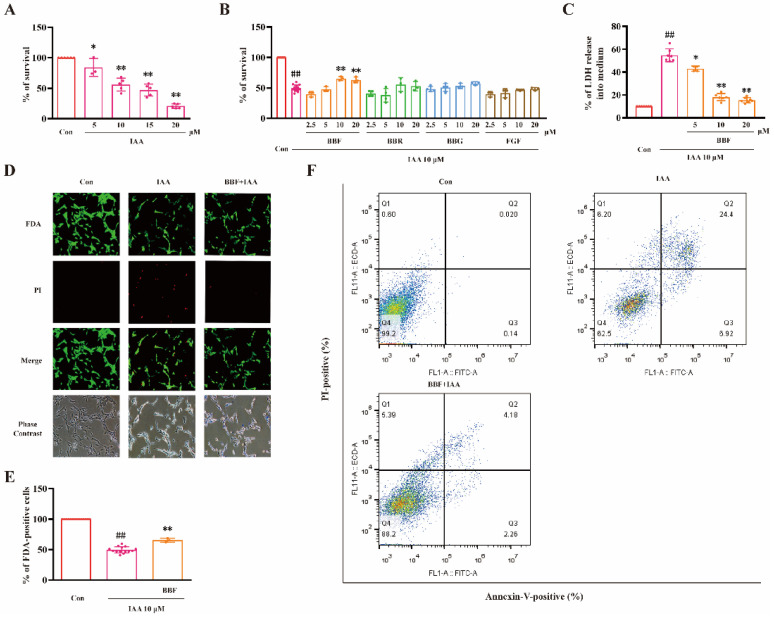
BBF but not its derivatives prevent chemical hypoxia-induced cell death in HT22 hippocampal cell line. (**A**) HT22 cells were exposed to various concentrations of IAA as indicated. Cell viability was measured by the MTT assay at 24 h after IAA exposure. (**B**) HT22 cells were pre-treated with BBF, BBR, BBG, or FGF at the indicated concentrations for 0.5 h, and then exposed to 10 μM IAA. Cell viability was measured by the MTT assay at 24 h after IAA exposure. (**C**–**E**) HT22 cells were pre-treated with BBF for 0.5 h, then exposed to 10 μM IAA. (**C**) The LDH release and (**D**) FDA/PI double staining was performed at 24 h after IAA exposure. (**E**) The percentage of FDA-positive cells was analyzed from representative photos. (**F**) HT22 cells was analyzed by flow cytometry at 24 h after IAA exposure, and the percentage of cells with live (Q4), early apoptosis (Q3), late apoptosis (Q2), and necrosis (Q1) conditions were pointed. Data represent the mean ± SEM; * *p* < 0.05 and ** *p* < 0.01 vs. the control group in (**A**); ## *p* < 0.01 vs. the control group; * *p* < 0.05 and ** *p* < 0.01 vs. IAA group in (**B**,**C**) and (**E**), (one-way ANOVA and Tukey’s test). Con: control; BBF: Brilliant Blue FCF; BBR: Brilliant Blue R; BBG: Brilliant Blue G; FGF: Fast Green FCF; IAA: Iodoacetic acid; LDH: Lactate dehydrogenase; FDA, Fluorescein diacetate; PI, Propidium iodide.

**Figure 2 pharmaceuticals-15-01018-f002:**
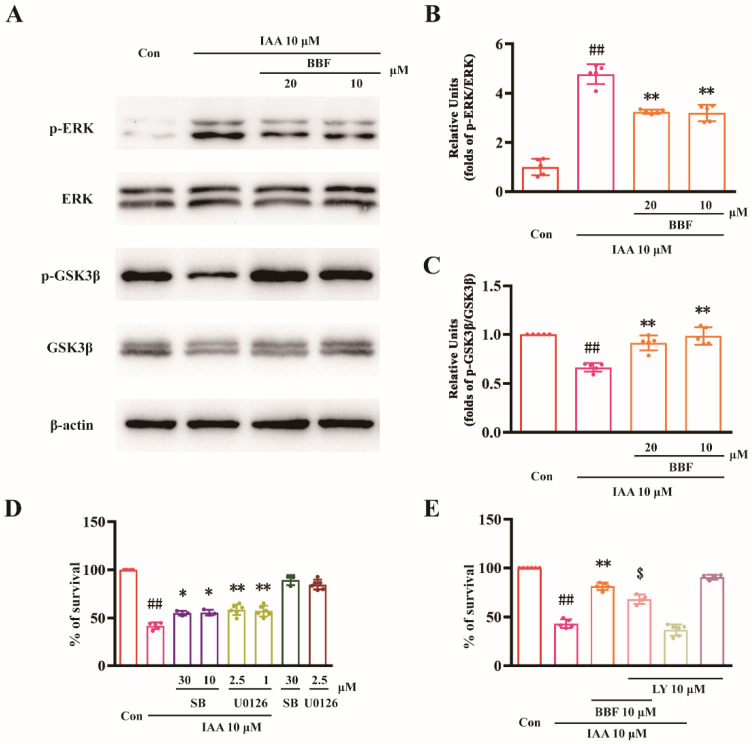
BBF produces anti-chemical hypoxia protective effects via the inhibition of ERK and GSK3β concurrently in HT22 cells. (**A**–**C**) HT22 cells were pre-treated with 10 and 20 μM BBF or vehicle control for 0.5 h, and then exposed to 10 μM IAA. Western blotting analysis was performed at 0.5 h after IAA exposure. (**A**) The representative blots were shown. The relative units of (**B**) p-ERK and (**C**) pSer9-GSK3β were demonstrated. (**D**) HT22 cells were pre-treated with SB415286, an inhibitor of GSK3β, or U0126, an inhibitor of MEK, at the indicated concentrations for 0.5 h, and then exposed to 10 μM IAA. Cell viability was measured by the MTT assay 24 h after IAA challenge. (**E**) HT22 cells were pre-treated with 10 μM LY294002 for 0.5 h, and then supplemented with 10 μM BBF for 0.5 h before the exposure to 10 μM IAA. Cell viability was then measured by the MTT assay at 24 h after IAA challenge. Data represent the mean ± SEM; ## *p* < 0.01 vs. the control group, * *p* < 0.05 and ** *p* < 0.01 vs. IAA group, $ *p* < 0.05 vs. BBF + IAA group (one-way ANOVA and Tukey’s test). ERK: extracellular regulated protein kinase; GSK3β: glycogen synthase kinase-3β; Con: control; BBF: Brilliant Blue FCF; IAA: Iodoacetic acid; LY: LY294002; SB: SB415286.

**Figure 3 pharmaceuticals-15-01018-f003:**
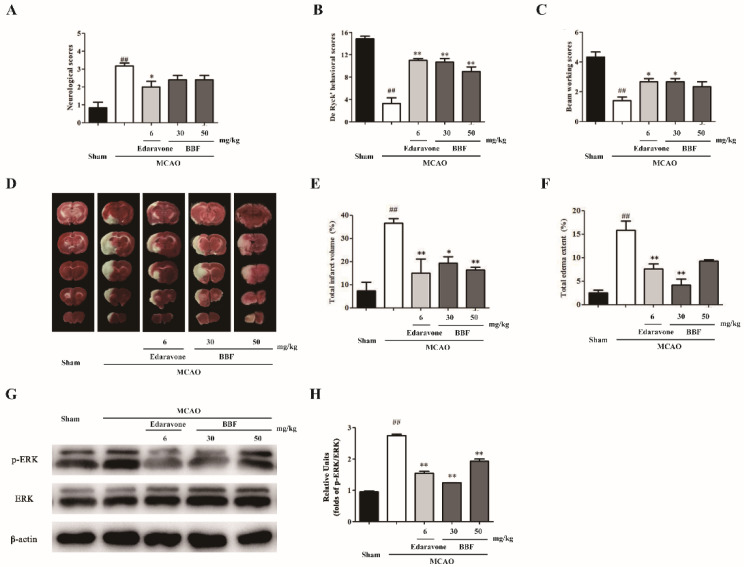
BBF prevents neurological and behavioral abnormity, brain infarct and ERK activation in MCAO rats. (**A**–**C**) Rats were injected with various drugs as indicated 30 min before, during and 30 min after MCAO surgery. The neurological and behavioral tests were measured 24 h after MCAO. (**A**) Neurological scores, (**B**) De Ryck’s behavioral scores, and (**C**) beam working scores were demonstrated. (**D**–**F**) Animals were sacrificed at 24 h after MCAO surgery. (**D**) The representative TTC staining images were demonstrated. Quantitative analysis for (**E**) infarct volume and (**F**) edema extent of each group was shown. (**G**) The representative blots were shown. (**H**) The relative units of p-ERK were demonstrated. Data represent the mean ± SEM (*n* = 6 in (**A**–**F**) and *n* = 3 in (**H**)); ## *p* < 0.01 vs. the Sham group, * *p* < 0.05 and ** *p* < 0.01 vs. MCAO group (one-way ANOVA and Tukey’s test). MCAO: middle cerebral artery occlusion/reperfusion; ERK: extracellular regulated protein kinase; TTC: 2,3,5-triphenyltetrazolium chloride; BBF: Brilliant Blue FCF.

**Figure 4 pharmaceuticals-15-01018-f004:**
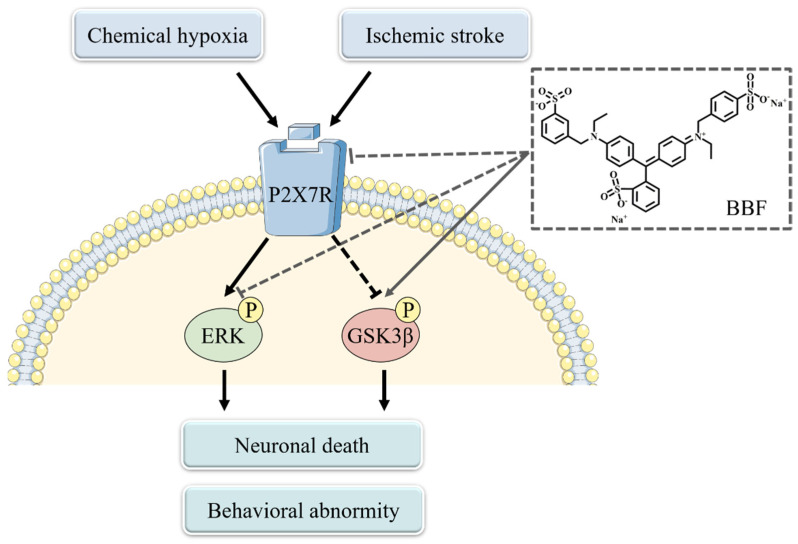
BBF prevents neuronal death and behavioral abnormity caused by chemical hypoxia and ischemic stroke. Chemical hypoxia and ischemic stroke activate P2X7 receptor, leading to activation of the GSK3β and ERK pathways, causing neuronal death and behavioral abnormity. BBF might produce neuroprotective effects via acting on ERK and GSK3β, possibly through the inhibition of P2X7 receptor. ERK: extracellular regulated protein kinase; GSK3β: glycogen synthase kinase-3β; BBF: Brilliant Blue FCF.

## Data Availability

Data is contained within the article.

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
