# Peer review of "Neuroprotective Effects of an Edible Pigment Brilliant Blue FCF against Behavioral Abnormity in MCAO Rats"

_pharmaceuticals, 2022, doi:10.3390/ph15081018_

Round 1

Reviewer 1 Report

The manuscript entitled “Neuroprotective effects of an edible pigment Brilliant Blue FCF against behavioral abnormity in MCAO rats” addresses the beneficial effect of  Brilliant blue FCF (BBF) in vitro in iodoacetate (IAA)-triggered neurotoxicity in HT22 cells and in vivo in middle cerebral artery occlusion/reperfusion rat model. The authors also explored the associated molecular mechanisms. Initially, the authors have proved that the BBF attenuates IAA-induced neurotoxicity via inhibition of ERK and GSK3-beta activation. In MCAO model, BBF mitigated the behavioral abnormality and inhibited ERK activation.  

The manuscript is clearly written, and the current findings are interesting.

Comments:     

1) In section 4.6. (Animals and surgery), how did the authors decide on the dose of the BBF (30 and 50 mg/Kg) in rats? How is the dose relevant to the human dose using the Human effective dose (HED) formula= animal dose x animal Km/ human Km (Nair AB, Jacob S. A simple practice guide for dose conversion between animals and humans. J Basic Clin Pharm. 2016 Mar;7(2):27-31). In the material and methods section, please also provide proper citations for selecting such doses.

2) What is the LD50 of BBF in rats? Are the used doses safe?

3) Why did the authors choose the intraperitoneal route for BBF administration? In fact, it is more commonly administered by humans via the oral route. Kindly, discuss the rationale, and please also provide the rationale for the timing and frequency of administration. Authors are advised to address this point and add the answers to the comment in the material and methods section.

4) To make it clearer for readers, the authors are advised to discuss the rationale for why edaravone was selected as the positive control. How is the dose of 6 mg/kg relevant to the human condition using the Human effective dose (HED) formula= animal dose x animal Km/ human Km (Nair AB, Jacob S. A simple practice guide for dose conversion between animals and humans. J Basic Clin Pharm. 2016 Mar;7(2):27-31). Authors are advised to address this point with proper citation and add the answers to the comment in the material and methods section.

5) Why did the authors choose the protective regimen of BBF for the current set of experiments? Pretreatment might not be relevant to the clinical case. Why didn't the authors try the therapeutic approach (i.e., administration of BBF after MCAO surgery) instead of the prophylactic regimen?

6) In the statistical analysis section, did the authors check data normality and homogeneity before proceeding to one-way ANOVA?

7) The authors are advised to add the cat no. for all kits and antibodies used in the current study.

8) As mandated by Pharmaceuticals journal, the authors are advised to incorporate the original uncropped Western blots as supplementary material to the current study files.

9) To make all figure legends stand-alone, authors are advised to add the full name of the used abbreviations at the end of each legend.

10) In figure 2, regarding the quantification of p-ERK and pSer9-GSK3β, please indicate the number of replicates for these Western blotting target proteins; were the replicates of WB derived from independent samples per group?

11) In figure 2D, why did not the authors try to show the effect of BBF treatment in presence of SB and U0126 inhibitors (IAA+BBF+ SB or U-126) similar to why they did for LY in figure 2E (IAA+BBF+ LY).

12) The quality of pics and graphs used in figure 3 needs to be enhanced. Please, provide higher resolution pics.

13) In figure 3, why did not the authors show the effect of BBF on pSer9-GSK3β/total GSK3β protein expression similar to what they did for p-ERK/total ERK protein expression?

14) To avoid confusion among readers on why the phosphorylation of GSK-3 beta was lowered in response to IAA treatment, the authors need to describe that pSer9-GSK3β is the inactive form of GSK3β. This is important in the sense that the authors have already described in the introduction section that “GSK3β is a multifunctional serine/threonine kinase acting on biological processes, such as neuronal apoptosis and neurotoxicity [11]”. Authors are advised to address this point and add the answers to the comment in the discussion section.

Author Response

1) In section 4.6. (Animals and surgery), how did the authors decide on the dose of the BBF (30 and 50 mg/Kg) in rats? How is the dose relevant to the human dose using the Human effective dose (HED) formula= animal dose x animal Km/ human Km (Nair AB, Jacob S. A simple practice guide for dose conversion between animals and humans. J Basic Clin Pharm. 2016 Mar;7(2):27-31). In the material and methods section, please also provide proper citations for selecting such doses.

Response:

Thank you for your suggestion. We noticed that the Joint FAO/WHO Expert Committee on Food Additives (JECFA) and the European Union Scientific Committee on Food (SCF) evaluated the bright blue food additive in 1970 and 1975, respectively, and an allowable daily intake (ADI) dose of 12.5 mg/kg bw/day were evaluated. In 1984, the ADI dose of BBF was revised to 10 mg/kg bw/day [1]. According to the Human effective dose (HED) formula, the dose of BBF at 61 mg/kg is equivalent to 10 mg/kg in human as a recommend ADI [2]. Therefore, we selected the doses of 30 and 50 mg/kg, which were slightly lower than 61 mg/kg of BBF to treat rats. We have accordingly added this information and provided the citations in our revised manuscript.

2) What is the LD50 of BBF in rats? Are the used doses safe?

Response:

Although the LD50 of BBF was not determined in rats, it was reported that daily administration of 631 mg/kg BBF was safe to rats in the chronic toxicity study [3]. The doses of BBF used in our study (30-50 mg/kg) were largely lower than 631 mg/kg. Therefore, the used doses of BBF is predicted to be safe to animals.

3) Why did the authors choose the intraperitoneal route for BBF administration? In fact, it is more commonly administered by humans via the oral route. Kindly, discuss the rationale, and please also provide the rationale for the timing and frequency of administration. Authors are advised to address this point and add the answers to the comment in the material and methods section.

Response:

Thank you for your suggestion. Oral administration is a convenience route for taking most of drugs in human. However, BBF is poorly absorbed from the gastrointestinal tract and mainly excreted unchanged [1]. Therefore, we chose intraperitoneal route for BBF administration. In addition, the timing for the treatment of ischemic stroke was limited, normally at the first several hours after the onset of the disease [4]. Therefore, rats were injected with BBF 30 min before, during and 30 min after MCAO surgery, to show the neuroprotection of BBF against ischemic stroke. We have accordingly added this information in the material and methods section of the revised manuscript.

4) To make it clearer for readers, the authors are advised to discuss the rationale for why edaravone was selected as the positive control. How is the dose of 6 mg/kg relevant to the human condition using the Human effective dose (HED) formula= animal dose x animal Km/ human Km (Nair AB, Jacob S. A simple practice guide for dose conversion between animals and humans. J Basic Clin Pharm. 2016 Mar;7(2):27-31). Authors are advised to address this point with proper citation and add the answers to the comment in the material and methods section.

Response:

Thank you for your suggestion. Edaravone is a clinical used drug for treating ischemic stroke and showed potent neuroprotection when used in the acute stage of ischemic stroke [5]. In the clinical study, edaravone was used at 1 mg/kg bw/day with a good safety profile with no adverse effect [6]. According to the HED formula, 6 mg/kg edaravone in rats is equivalent to 1 mg/kg edaravone in human. Therefore, we selected edaravone at 6 mg/kg as a positive control drug in the study. We have accordingly added this information in the material and methods section of revised manuscript.

5) Why did the authors choose the protective regimen of BBF for the current set of experiments? Pretreatment might not be relevant to the clinical case. Why didn't the authors try the therapeutic approach (i.e., administration of BBF after MCAO surgery) instead of the prophylactic regimen?

Response:

Thank you for your critical comment. Indeed, it is encouraged to show the therapeutic effects rather than the protective effects for anti-stroke candidates. However, before the investigation of the therapeutic effects of BBF, we needed to explore if this compound could really produce neuroprotection against ischemic stroke. In this study, we mainly described the possibility of BBF to prevent ischemic stroke. And the therapeutic effects of BBF against MCAO-induced neurological impairments are being investigated in our lab.

6) In the statistical analysis section, did the authors check data normality and homogeneity before proceeding to one-way ANOVA?

Response:

Thank you for your suggestion. We have checked the normality and homogeneity of the data.

7) The authors are advised to add the cat no. for all kits and antibodies used in the current study.

Response:

We have accordingly added Cat No. for kits and antibodies in our revised manuscript.

8) As mandated by Pharmaceuticals journal, the authors are advised to incorporate the original uncropped Western blots as supplementary material to the current study files.

Response:

We have provided most of original images of Western blots as a supplemental figure in our revised manuscript. However, some of original images were missed in our lab, and we are currently performing the experiments, and will replace these Western blots by new images. We have asked for one-month extension for performing such experiments.

9) To make all figure legends stand-alone, authors are advised to add the full name of the used abbreviations at the end of each legend.

Response:

We have accordingly added abbreviations at the end of each legend in our revised manuscript.

10) In figure 2, regarding the quantification of p-ERK and pSer9-GSK3β, please indicate the number of replicates for these Western blotting target proteins; were the replicates of WB derived from independent samples per group?

Response:

We repeated 3 times for Western blotting and the replicates of WB were derived from independent samples.

11) In figure 2D, why did not the authors try to show the effect of BBF treatment in presence of SB and U0126 inhibitors (IAA+BBF+ SB or U-126) similar to why they did for LY in figure 2E (IAA+BBF+ LY).

Response:

LY294002 (LY) in a PI3-K inhibitor and abolished BBF-induced activation of Akt pathway. Therefore, the comparison of cell viability in IAA+BBF+LY and IAA+BBF groups demonstrated the neuroprotection of BBF was directly from the activation of Akt pathway. SB415286 (SB) is GSK3β inhibitor and U0126 is a MEK inhibitor. The application of SB and U0126 could mimic the actions of BBF on Akt and ERK pathways, respectively. By comparing the cell viability in IAA+SB+U0126 and IAA groups, it is concluded that the activation of Akt and the inhibition of the ERK pathway induced neuroprotection in IAA-treated HT22 cells. BBF was proved to activate the Akt pathway and to inhibit the ERK pathway, simultaneously. These results also indicated that BBF produced neuroprotection against IAA-induced neurotoxicity via the activation of the Akt pathway and the inhibition of ERK pathways.

12) The quality of pics and graphs used in figure 3 needs to be enhanced. Please, provide higher resolution pics.

Response:

We have accordingly enhanced the quality of all pictures and graphs in our revised manuscript.

13) In figure 3, why did not the authors show the effect of BBF on pSer9-GSK3β/total GSK3β protein expression similar to what they did for p-ERK/total ERK protein expression?

Response:

Our results showed that pSer9-GSK3β was not changed in the study. Therefore, we did not add this figure in our manuscript.  

14) To avoid confusion among readers on why the phosphorylation of GSK-3 beta was lowered in response to IAA treatment, the authors need to describe that pSer9-GSK3β is the inactive form of GSK3β. This is important in the sense that the authors have already described in the introduction section that “GSK3β is a multifunctional serine/threonine kinase acting on biological processes, such as neuronal apoptosis and neurotoxicity [11]”. Authors are advised to address this point and add the answers to the comment in the discussion section.

Response:

Thank you for your suggestion. We have accordingly added this information in our revised manuscript.

Reviewer 2 Report

Congratulations to the authors for the amount of fine work aimed to contribute to a major health problem of wide interest.

Author Response

1) I suggest to mention here the chemical nature of the pigmen.

Response:

We have accordingly added the chemical nature of the pigment in our revised manuscript.

2) I suggest to mention some of such side effects

Response:

We have accordingly revised in our manuscript.

3) I suggest to rephrase "And the blockade of GSK3β has been shown to produce neuroprotective effects against ischemic stroke [12]. " to "However, the blockade of GSK3β has been shown to produce neuroprotective effects against ischemic stroke [12]. "

Response:

We have accordingly revised in our manuscript.

4) I suggest to mention which derivatives.

Response:

We have accordingly revised in our manuscript.

5) Add “with the purpose of...”

Response:

We have accordingly revised in our manuscript.

6) "HT22 ells were" correct to HT22 cells were

Response:

We have accordingly revised in our manuscript.

7) Suggestion to change "could not produce " to "did not exert" or "did not show"

Response:

We have accordingly revised in our manuscript.

8) Figure 1: A high resolution image is strongly suggested. It will improve greatly the manuscript.

Response:

We have replaced the figure with a high-resolution image in the revised manuscript.

9) Figure 2: A high resolution image will improve visualizing the graph material. This applies to all graphs within the manuscript

Response:

We have replaced the figure with a high-resolution image in the revised manuscript.

10) A reference to source or chemical structure of edaravone should be made

Response:

We have accordingly revised in our manuscript.

11) Figure 4: Please reconstruct the chemical formula including the charge on atoms. A thicker bond setting will improve visual appreciation

Response:

We will submit the original documents in the attachment.

12) The medium was replaced....

Response:

We have accordingly revised in our manuscript.